# Antigenic Characterization of Infectious Bronchitis Virus in the South China during 2021–2022

**DOI:** 10.3390/v15061273

**Published:** 2023-05-29

**Authors:** Weifeng Yuan, Ting Lv, Weiwei Jiang, Yuechi Hou, Qingyi Wang, Jinlian Ren, Lei Fan, Bin Xiang, Qiuyan Lin, Chan Ding, Tao Ren, Libin Chen

**Affiliations:** 1College of Veterinary Medicine, South China Agricultural University, Guangzhou 510642, China; 2National and Regional Joint Engineering Laboratory for Medicament of Zoonosis Prevention and Control, Guangzhou 510642, China; 3Key Laboratory of Animal Vaccine Development, Ministry of Agriculture, Guangzhou 510642, China; 4Key Laboratory of Zoonosis Prevention and Control of Guangdong Province, Guangzhou 510642, China; 5College of Veterinary Medicine, Yunnan Agricultural University, Kunming 650201, China; xiangbin2018@126.com; 6Shanghai Veterinary Research Institute (SHVRI), Chinese Academy of Agricultural Sciences (CAAS), Shanghai 200241, China

**Keywords:** infectious bronchitis virus, chicken coronavirus, antigenic characterization, molecular evolution

## Abstract

Avian infectious bronchitis is a serious and highly contagious disease that is caused by the infectious bronchitis virus (IBV). From January 2021 to June 2022, 1008 chicken tissue samples were collected from various regions of southern China, and 15 strains of the IBV were isolated. Phylogenetic analysis revealed that the strains mainly comprised the QX type, belonging to the same genotype as the currently prevalent LX4 type, and identified four recombination events in the S1 gene, among which lineages GI-13 and GI-19 were most frequently involved in recombination. Further study of seven selected isolates revealed that they caused respiratory symptoms, including coughing, sneezing, nasal discharge, and tracheal sounds, accompanied by depression. Inoculation of chicken embryos with the seven isolates resulted in symptoms such as curling, weakness, and bleeding. Immunization of specific pathogen-free (SPF) chickens with inactivated isolates produced high antibody levels that neutralized the corresponding strains; however, antibodies produced by vaccine strains were not effective in neutralizing the isolates. No unambiguous association was found between IBV genotypes and serotypes. In summary, a new trend in IBV prevalence has emerged in southern China, and currently available vaccines do not provide protection against the prevalent IBV strains in this region, facilitating the continued spread of IBV.

## 1. Introduction

Avian infectious bronchitis (IB) is caused by the avian infectious bronchitis virus (IBV) and is a significant economic threat to poultry farms. Although the IBV has a wide range of hosts, galli and pheasants are generally believed to be its natural hosts. In chickens, IBV infection primarily causes respiratory tract infections, nephritis, and decreased productivity [1]. In layer hens, infection can result not only in clinical symptoms but also in permanent and irreversible damage to the reproductive system, resulting in a loss of maximum egg production and a reduction in the quality of eggs [2]. Some IBV strains can cause intestinal, glandular, and muscular diseases. IB is one of the most significant infectious diseases affecting the poultry industry worldwide. Currently, vaccines remain the most critical measure for prevention and control, with live-attenuated vaccines being the most widely used [3].

The IBV belongs to the *Gammacoronavirus* genus of the *Coronaviridae* family in the order Nidovirales [4] The IBV genome encodes four structural proteins: nucleocapsid (N) protein, envelope (E) protein, membrane (M) glycoprotein, and spike (S) glycoprotein [5]. The S protein is the major structural protein of the IBV and is cleaved into S1 and S2 subunits upon translation, facilitating the binding of the virus to its receptor and the subsequent entry into host cells [6]. The S1 protein contains an independent receptor-binding domain that induces neutralizing antibodies, inhibits blood clotting, and generates epitopes for serotype-specific antibodies [7]. Like SARS-CoV-2, IBV replication depends on RNA polymerases and lacks proofreading capabilities, making it prone to mutations or recombination during replication [8]. As the pressure on antibodies induced by IBV vaccines increases, the genetic variation in the IBV is accelerated, leading to the emergence of new genotypes and serotypes [9]. Insertions, deletions, point mutations, and recombinations in the S1 nucleotide sequence can directly lead to the emergence of several serotypes/genotypes. Differences in S1 proteins among the different strains have resulted in weakened cross-protection among the strains [10]; therefore, analysis of the nucleotide sequence encoded by S1 has conventionally been used to determine the genetic type of IBV [11]. Among the numerous classifications based on the S1 gene, the most commonly used classification currently divides the IBV into at least 35 lineages, including seven genotypes (GI-GVII) with many additional inter-lineage recombinants [12].

The IBV has been highly prevalent in China for a long time, with a wide geographical range of infections that has been harmful to the development of the poultry industry. Since its first isolation in 1996, the lineage GI-19 (QX-type) has become the most prevalent IBV strain in China in the last decade [13]. Moreover, lineages GI-7 (TW-type) and GI-13 (4/91-type) have also been reported as major IBV strains in China, with lineages GI-1 (Mass-type), GI-9 (Ark-type), and GI-28 (LDT3-type) sporadically reported [14,15]. The isolation rate of lineage GVI-1 IBV strains has also increased in recent years [16,17], as part of the critical continuous epidemiological surveillance of the complex prevalence of the IBV in China.

In this study, we conducted epidemiological surveillance of the IBV in China from January 2021 to June 2022. We identified the S1 gene in 15 IBV strains isolated from clinical samples and investigated its prevalence, genotype, and recombination. Additionally, we explored the antigenic characteristics of the isolated strains and investigated the relationship between genotype and serotype.

## 2. Materials and Methods

### 2.1. Virus Isolation and Identification

Tissue samples of the trachea, lungs, kidneys, spleens, intestines, and joint effusion were collected from deceased chickens suspected of IBV infection across 168 chicken farms located in the Yunnan, Guizhou, Guangxi, and Guangdong provinces of China in 2021–2022. All sick chickens were immunized with H120 or 4/91. The samples were frozen and thawed three times, treated with PBS containing 200 U/mL penicillin and 200 µg/mL streptomycin, then centrifuged at 7000× *g* for 5 min. After maintaining the samples at 4 °C for 3 h, the viruses were propagated three times by blind passaging. In each blind passage, 9-day-old embryos of specific pathogen-free (SPF) chickens were inoculated with 0.2 mL of the supernatant of each isolate via the allantoic cavity. The allantoic fluids were harvested after incubation at 37 °C for 48 h post-inoculation. RT-PCR was used to identify the presence of the IBV and samples infected with other viruses were excluded to ensure the presence of only the IBV. The primers used for RT-PCR are listed in Appendix A.

### 2.2. Cloning and Sequencing of S1

According to the IBV sequence published in GenBank, the primer pair of IBV-S1 was used to amplify S1. The HiScript II One Step RT-PCR Kit (Vazyme, Nanjing, China) instructions were followed, using the total RNA extracted from the virus obtained from the chicken embryo allantoic fluid as the template. RT-PCR involved reverse transcription at 50 °C for 30 min, followed by 94 °C for 3 min, 30 cycles of denaturation at 94 °C for 10 s, annealing at 54 °C for 30 s, and extension at 72 °C for 30 s, with a final extension at 72 °C for 5 min. The RT-PCR products were approximately 1743 bp and were detected by electrophoresis on a 1.0% agarose gel, then observed using an ultraviolet transilluminator. PCR amplicons were purified from the agarose gels using the Gel DNA Extraction Mini Kit (Vazyme, Nanjing, China) according to the manufacturer’s instructions. Purified PCR products were then inserted into the pMD19-T vector (Takara, Beijing, China) and transformed into DH5α Escherichia coli competent cells. This was followed by the sequencing of the plasmid inserts.

### 2.3. Phylogenetic Analysis

In addition to the IBVs isolated in this study, 107 S1 gene sequences of 35 lineage IBV strains were obtained from GenBank (NCBI, Bethesda, MD, USA). All sequences were aligned using MAFFT v7.221.3 [18]. A maximum likelihood (ML) tree was constructed based on the Sl gene sequences and GTR+F+R5 model, selected with Modelfinder [19], using IQ-TREE software [20] with 1000 bootstrap replicates. The ML tree was visualized using Figtree version 1.4.4 (http://tree.bio.ed.ac.uk/software/figtree/ (accessed on 2 May 2023)).

### 2.4. Recombination Analysis

Putative recombination events and parental strains were identified using the Recombination Detection Program version 4.0 (RDP 4.0; Simmonics, University of Warwick, Coventry, UK). Multiple methods and default program settings were used to analyze the data, including RDP Bootscan, GeneCony, Maxchi, Chimaera, SiSscan, LARD, 3Seq, and PhylPro [21]. The most likely recombinant fragments (*p* ≤ 10^−12^) were determined along with the possible parental virus and the beginning and end points. The potential recombination events and breakpoints were determined by similarity plot (SimPlots) analysis in SimPlot version 3.5.1, using a window of 200 bp and a step size of 20 bp.

### 2.5. Inactivated IBV Vaccine and Testing

Twenty-eight-day-old SPF chickens were purchased from Guangdong Wens Dahuanong Biotechnology Co., Ltd. (Yunfu, China) and fed in a negative-pressure biosafety isolation chamber. The chickens had free access to water and commercial feed. The immunization and virus challenge procedures are shown in Figure 1. We used ultracentrifugation to concentrate the virus to achieve an EID_50_ value higher than 5.5. Viral stocks were inactivated with the addition of formaldehyde (Sigma, Shanghai, China) to 0.2%. The formaldehyde inactivated antigen solution was emulsified with oil adjuvant (ISA78VG, Seppic, Paris, France) at a ratio of 25:75 (*w*/*w*). The chickens were immunized with the oil-emulsion vaccine using 10^5.5^ EID_50_ in 0.2 mL per chicken. Vaccination and challenge studies used the 28-day-old SPF chickens in 9 groups of 5 chickens each. Groups 1 to 7 were vaccinated intramuscularly with 200 µL IBV isolate strains of oil-emulsion vaccine containing 10^5.5^ EID_50_ antigens per chicken. Groups 8 and 9 were vaccinated intramuscularly with 200 µL IBV H120 or 4/91 strains of oil-emulsion vaccine containing 10^5.5^ EID_50_ antigens per chicken. Groups 10 to 16 were inoculated with sterile PBS and used as infection control that was unvaccinated but challenged. Booster vaccinations were carried out after 1 and 2 w, and the chickens were challenged with 0.2 mL of IBV isolate or vaccine strains containing 10^5.5^ EID_50_ virus. Serum samples were obtained before each immunization and IBV challenge, then tested by ELISA (IDEXX 99-09262, Westbrook, ME, USA) and neutralization tests.

### 2.6. Antigenic Correlation Coefficient

The correlation coefficient (R) = √(r1×r2), where r1 = heterologous serum titer 1/homologous serum titer 1 and r2 = heterologous serum titer 2/homologous serum titer 2. When R > 80%, it was considered the same serotype. When R was between 25% and 80%, it was considered a different subtype of the same serotype. When R ≤ 25%, it was considered a different serotype.

## 3. Results

### 3.1. Epidemic Information and S1 Sequences of Clinical Samples

Between January 2021 and June 2022, 1008 samples were collected from broiler farms in southern China. After excluding the samples that were co-infected with the IBV and other viruses, a total of 15 IBV strains were isolated and identified, including 11 isolates from Guangxi, two from Yunnan, and two from Guangdong province. The accession numbers and the clinical information for each strain are summarized in Table 1.

The 15 isolates of the IBV exhibited different clinical symptoms, of which eight strains (53.33%) were from chickens with typical respiratory clinical symptoms, including wheezing, coughing, sneezing, tracheal rales, and nasal discharge, and autopsy showed hemorrhagic points in the lungs and trachea; three strains (20%) were from chickens with typical intestinal symptoms, such as severe dehydration and weight loss, and autopsy showed hemorrhagic points in the intestines; three strains (20%) were from chickens with typical joint inflammation symptoms, with swollen and fluid-filled foot joints visible; and one strain (6.67%) was from chickens with typical renal symptoms, such as kidney enlargement and urate-filled ureters, and autopsy showed hemorrhagic points in the kidneys.

The nucleic acid sequences, amino acid sequence lengths, and cleavage sites of S1 in the 15 IBV isolates were analyzed. Within S1, there were four different nucleotide (1611, 1617, 1620, and 1626) and deduced amino acid (537, 539, 540, and 542) sequence lengths. The most common S1 was 1620 nucleotides long, accounting for 60% of the total IBV isolates. There were five types of cleavage sites, including HRRRR, RRFRR, SRRLR, and RRSRR, all of which were similar to those reported previously (Feng et al., 2014).

### 3.2. Evolutionary Genetic and Recombination Analysis of S1

A phylogenetic tree of S1, including 107 reference strains and the 15 IBV isolates, was constructed using IQ-TREE. As shown in Figure 2, the 15 IBV isolates comprised four main genotypes: GI-19 (QX), GI-13 (4/91), GI-1 (Mass), and GI-7 (TW-I), with GI-19 (QX) being the predominant genotype. Two isolates, 220149YNKM and 210059GXYL, formed independent evolutionary branches and were considered variant strains.

Based on different genotypes, the 15 IBV isolates were aligned and compared with the H120 and 4/91 vaccine strains. The results showed that the nucleotide and amino acid sequence similarities were 77 to 99.9% and 74.4 to 99.6%, respectively (Table 2). The isolates of each genotype showed the highest similarity to the corresponding vaccine strain. GI-7 (TW-I) did not have any equivalent vaccine strains and the similarity of S1 with the two major vaccine strains was low.

Recombination analysis of the S1 gene sequences of all 15 IBV strains isolated in this study was performed using RDP 4.0. Four isolates, 210059GXYL, 210099GXNN, 220149YNKM, and 220198GDZC, exhibited recombination events (Figure 3). The breakpoint positions and specific P-values for each recombination event detection method are listed in Appendix A. As shown in Figure 3, recombination of multiple IBV lineages was observed, and lineage GI-13 and GI-19 IBVs were identified as the major and minor parental strains, respectively, in these four recombination events. Together, these results indicated that the prevalent IBV lineages had a higher probability of being involved in recombination events.

### 3.3. Antigenic Characterization of Isolate Strains

To investigate whether the recombinant strains had any impact on the virulence of the IBV, typical virulent strains were selected as representative branches for antigenic characterization studies, including four recombinant strains and three typical strains of GI-1, 13, and 19. After inoculating SPF chicken embryos with the isolates, all seven isolates caused embryonic lesions and death, together with various symptoms such as weak, curled, and dissolved embryos (Figure 4A,B).

The isolated IBV strains were inactivated and administered to SPF chickens for immunization. After the third immunization, the chickens were challenged with the corresponding strains and no typical IBV symptoms were observed in the immunized SPF chickens. In contrast, unimmunized chickens infected with the isolated strains exhibited symptoms such as depression, ruffled feathers, and coughing, consistent with the symptoms of the source-diseased chicken (Appendix A). Autopsy results confirmed this finding, as bleeding points in the trachea and lungs and typical IBV infection symptoms in the kidneys were only observed in the unimmunized group (Figure 4C–J). This further demonstrated the severe damage caused by IBV infection and the efficacy of immunization with the isolated strains in protecting against IBV infection.

ELISA revealed that the immunized chickens produced high levels of IBV antibodies, with the highest level reached after the third immunization, and had high neutralizing titers against the corresponding strains (Figure 5A). To confirm whether the recombinant strain had immune escape ability, cross-serum neutralization experiments were conducted using different groups of serum. The cross-protective antibodies generated by different genotypes of IBV strains had lower protection rates, and the serum antibodies produced by the H120 and 4/91 vaccine strains had poor neutralizing ability against the isolated strains. Only 32 had neutralizing ability against 220161YNKM and 210127GXYL, indicating that differences in serotypes among isolates of the prevalent IBV strains in the southern region of China may have developed an immune escape ability against vaccines (Figure 5B).

Further analysis of the antigenic correlation between the vaccine strains and isolated strains confirmed significant differences in antigenicity, with hemagglutination inhibition antigen correlation coefficients ranging from 2.21 to 74.56 (Table 3). The antigen correlation coefficient between different isolated and vaccine strains was as low as 1.05 and as high as 80.03 between 210059GXYL and 220149YNKM. Although 210085GXYL and 210099GXNN were both GI-19, and 210127GXYL and 4/91 were both GI-13, their R values were all less than 80, indicating that the genotypes and serotypes did not fully match. Therefore, the existing H120 and 4/91 vaccines may not effectively prevent or control IBV infection. It is essential to use appropriate vaccines to achieve ideal preventive and control effects.

## 4. Discussion

Coronaviruses comprise a large family and are widely found in nature. A growing body of research suggests that SARS-CoV-2, which caused a global pandemic in late 2019, may have originated in nature [22]. Therefore, the study of coronaviruses in different species has received increasing attention from the scientific community. Chicken-derived coronaviruses (IBVs) are widespread worldwide and cause significant losses and threats to the poultry industry. In this study, an epidemiological survey of IBV conducted in South China isolated 15 IBV strains from 1008 infected chicken tissue samples collected between January 2021 and June 2022. Compared with the analyses by Lian et al. (2021) in South China between April 2019 and March 2020 (139/420) and Xu et al. (2018) in China from January 2016 to December 2017 (213/801), the IBV isolation rate reported in the current study was significantly reduced [9,15]. A credible explanation for this result is that the clinical samples we collected included chickens that did not have respiratory diseases due to IBV infection. We also isolated other pathogens, including the avian influenza virus (AIV), the Newcastle disease virus (NDV), the adenovirus, and others, including AIV-mixed infections. As these pathogens may have caused the disappearance of the IBV during chicken embryo passage, we excluded these samples. Due to the varying levels of biosafety in aquaculture farms across different provinces and the significant differences in sample numbers, monitoring bias was also an inevitable challenge. These potential variables in collected samples will be addressed in future studies. Overall, our results showed that the GI-19 (QX-type) IBV lineage still dominated the epidemic. Other IBV lineages were sporadically isolated, similar to the results of recent research in China [9,23].

Recombination is an important contributing factor to the emergence and evolution of the IBV, as well as the emergence of new coronaviruses and novel diseases [12,24]. We identified four recombination events in the S1 gene, among which the IBVs from lineages GI-13 and GI-19 were more frequently involved in recombination, resulting in wider tissue tropism and leading to a more diverse range of clinical symptoms, including respiratory, pulmonary, renal, and even arthritic symptoms. Our findings indicate that these major prevalent lineages may play key roles in the transmission of the IBV in South China.

Ji et al. (2020) found that the homology between IBV variant strains in central China and vaccine strains was only 67.4–89.8% [23]. In this study, the nucleotide and amino acid sequence similarities were 77–99.9% and 74.4–99.6%, respectively, between isolated IBV strains and H120 or 4/91, indicating the limited effectiveness of the vaccine against the variant strains. The S1 protein has conformation-dependent epitopes that induce virus neutralization and serotype-specific antibodies. To investigate whether mutations in the S1 gene affect the protective efficacy of IBV vaccine strains, seven isolate strains were selected from genotype and isolation sites, among which 210059GXYL, 220198GDZC, 220149YNKM, and 210099GXNN were potential strains for S1 gene recombination, 210085GXYL was a GI-19 strain isolated from Guangxi, 210127GXYL was a GI-13 strain isolated from Guangxi, and 220161YNKM was the only GI-1 strain isolated from Yunnan. Then, we examined plasma samples obtained from SPF chickens immunized three times with inactivated isolates. The immunized SPF chickens were found to have high antibody levels and sufficient protective power against the corresponding strains. No clinical symptoms were observed in the immunized SPF chickens after challenge and the results of the serum neutralization experiments confirmed this conclusion. However, the H120 and 4/91 vaccines were unable to provide sufficient protection against the isolate, indicating that the recombinant strain may have developed an immune escape from the current vaccines, limiting their effectiveness in disease control. Li et al. (2022) conducted a survey of IBV prevalence in southwest China and observed the same phenomenon: IBV vaccines were unable to suppress virus release [25]. Therefore, effective vaccines targeting local strains and improved vaccination strategies must be developed.

The IB vaccines currently used in China are mainly targeted against MASS strains, such as H120, which differ in genotype from isolate strains, such as GI-19. Serotype is a reference for representing IBV antigenic variation. Although the nucleotide homology between the isolated and vaccine strains ranged from 77–99.9%, the antigenic correlation coefficient was low and the correlation between strains of the same genotype was less than 80%, indicating that the genotype and serotype did not match. Thus, S1 alone was insufficient to characterize the serotype or protectotype. Chen et al. (2015) analyzed five H120-recombinant strains and found that H120 did not protect against two IBV strains with a genotype- and serotype-matching rate of only 60% [26]. Using reverse genetics, Shan et al. (2018) further demonstrated that recombination of the S1 gene affects the IBV serotype [27]. Yan et al. (2021) also found that the ongoing evolution of IBV field strains through genetic recombination and mutation leads to outbreaks among vaccinated chicken populations, and suggested that evaluating the virulence of the virus strain based solely on the S1 is insufficient [28]. These studies indicated that antigenic variation in IBV cannot be measured solely by genotype; therefore, the selection of serotypes is an important reference index for IBV vaccine application, to conduct continuous epidemiological surveillance against the IBVs, and to screen for the most prevalent genotypes and serotypes to identify vaccine candidates for IB prevention and control in China.

In summary, we isolated 15 IBV strains from South China between 2021 and 2022, with nucleotide homology ranging from 77–99.9% and divided them into four genotypes, with GI-19 serving as the main genotype. Currently, vaccines are no longer effective in protecting against the prevalent IBV strains in southern China, as these strains have developed immune escape. Our study provides valuable information for the control of the IBV in southern China, which may help guide the design of vaccination strategies.

## Figures and Tables

**Figure 1 viruses-15-01273-f001:**
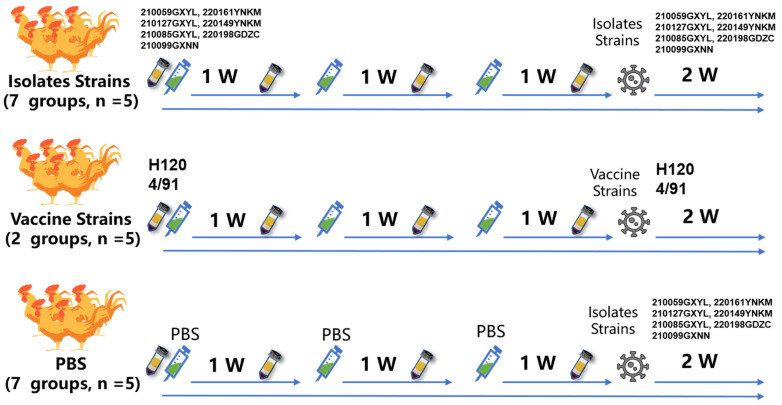
Grouping information and timing of immunization, virus challenge, and serum acquisition.

**Figure 2 viruses-15-01273-f002:**
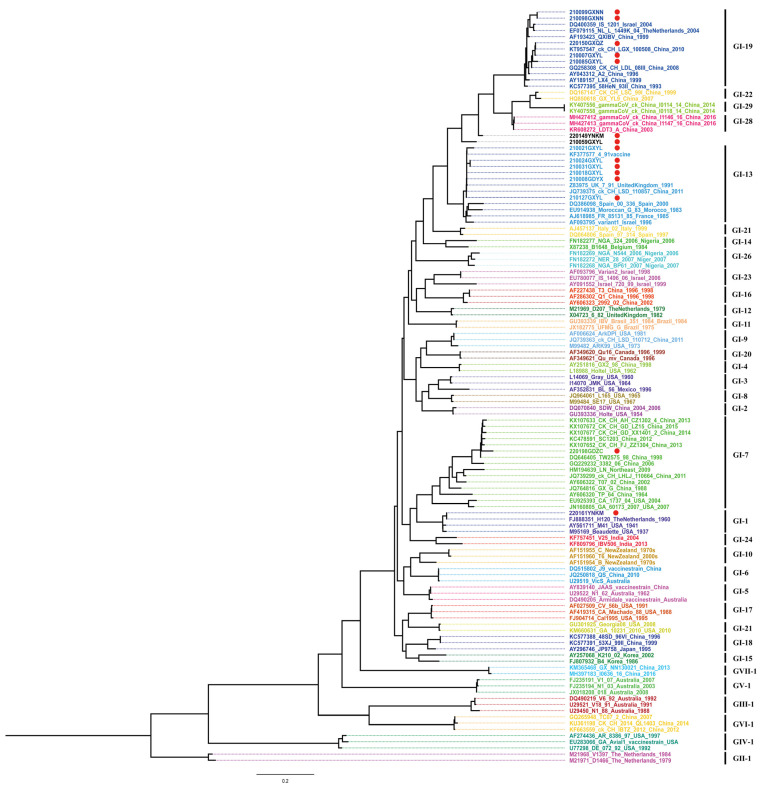
Maximum likelihood (ML) tree based on the S1 gene from 35 lineages of IBV. The tree was constructed using IQ-TREE software with the GTR+F+R5 model. Red circles indicate the stains isolated in this study.

**Figure 3 viruses-15-01273-f003:**
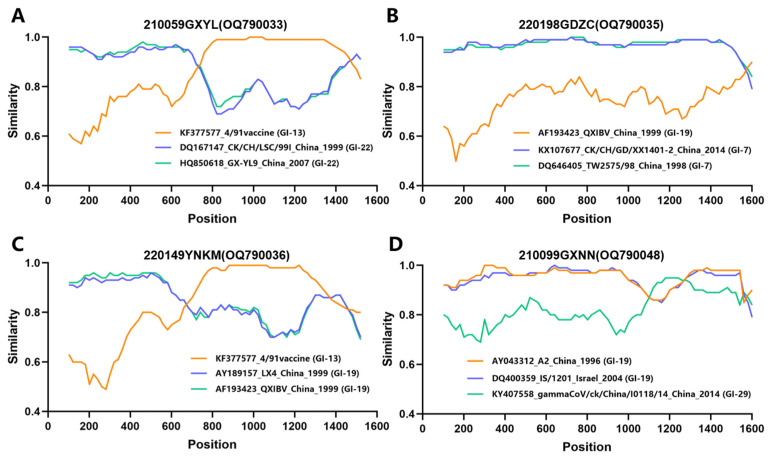
Recombination events in the S1 gene from the isolated IBV strains. Simplot analysis was performed to detect recombination within the S1 gene from (**A**) 210059GXYL, (**B**) 220198GDZC, (**C**) 220149YNKM, and (**D**) 210099GXNN. The *y*-axis represents the ratio of identity within a 200-bp wide sliding window centered on the position plotted, with a 20 bp step size between plots.

**Figure 4 viruses-15-01273-f004:**
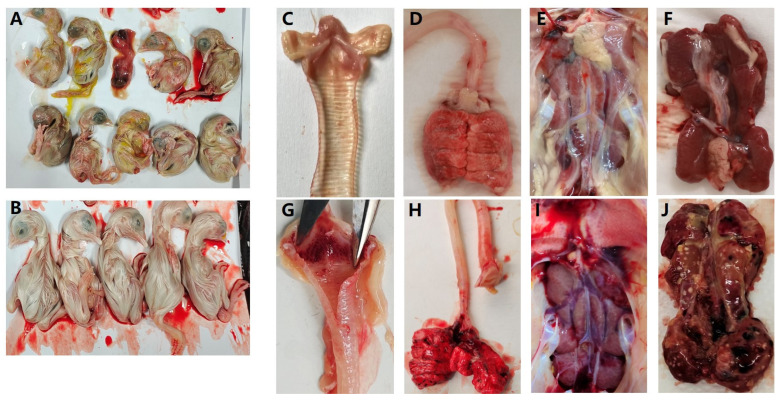
(**A**) SPF chicken embryos infected by IBV isolated strains; (**B**) control; (**C**–**F**) the trachea, lung, and kidney from immunizated chickens with isolated strains; (**G**–**J**) the trachea, lung, and kidney from immunizated chickens with PBS.

**Figure 5 viruses-15-01273-f005:**
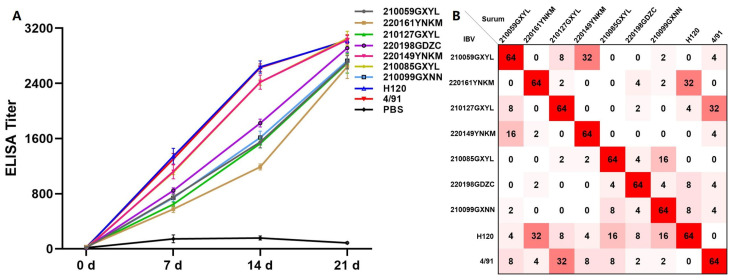
(**A**) Antibody response of chickens after immunization with isolated strains and vaccine strains; (**B**) cross-serum neutralization experiments between isolated strains and vaccine strains.

**Table 1 viruses-15-01273-t001:** Background and S1 information of domestic isolates of the IBV from 2021 to 2022.

IBV Isolates	Accession Number	Province ^1^	Date (Day-Month-Year)	Vaccination History	Clinical Type	Length of S1 (nt/aa) ^2^	Cleavage Recognition Motifs ^3^	Genotype
210007GXYL	OQ790027	Guangxi	05-08-2021	H120 + H120	A	1620/540	HRRRR	GI-19
210008GDYX	OQ790028	Guangdong	11-08-2021	H120 + H120	A	1617/539	RRSRR	GI-13
210018GXYL	OQ790029	Guangxi	06-09-2021	H120 + H120	RS	1617/539	RRSRR	GI-13
210021GXYL	OQ790030	Guangxi	06-09-2021	H120 + H120	I	1617/539	HRRRR	GI-13
210024GXYL	OQ790031	Guangxi	08-09-2021	H120 + H120	A	1620/540	RRSRR	GI-13
210031GXYL	OQ790032	Guangxi	16-09-2021	H120	RS	1620/540	RRSRR	GI-13
210059GXYL	OQ790033	Guangxi	29-10-2021	H120	RS	1626/542	SRRLR	—
220198GDZC	OQ790035	Guangdong	07-11-2021	H120	RS	1620/540	HRRRR	GI-7
220149YNKM	OQ790036	Yunnan	18-11-2021	H120 + H120	RS	1617/539	RRFRR	—
210085GXYL	OQ790037	Guangxi	03-11-2021	H120 + H120	I	1620/540	HRRRR	GI-19
210127GXYL	OQ790043	Guangxi	12-11-2021	4/91	I	1617/539	RRSRR	GI-13
220150GXQZ	OQ790044	Guangxi	19-01-2022	H120 + H120	NP	1620/540	HRRRR	GI-19
220161YNKM	OQ790045	Yunnan	10-03-2022	H120	RS	1611/537	RRFRR	GI-1
210098GXNN	OQ790047	Guangxi	07-11-2021	H120	RS	1620/540	HRRRR	GI-19
210099GXNN	OQ790048	Guangxi	07-11-2021	H120	RS	1620/540	HRRRR	GI-19

Abbreviations: IBV, infectious bronchitis virus; NP, nephropathogenic; RS, respiratory; A, Arthritis; I, Intestinal. 1: Province where the viruses were isolated; 2: Length of nucleotides and deduced amino acids; 3: Cleavage recognition motifs of S1; R, arginine; F, phenylalanine; H, histidine; T, threonine; K, lysine; L, leucine; S, serine.

**Table 2 viruses-15-01273-t002:** Similarity in nucleotides and amino acids between isolate strains and vaccine strains.

	aa (%)																		
nt (%)		1	2	3	4	5	6	7	8	9	10	11	12	13	14	15	16	17	
	1	***	78.7	78.9	80.4	80.6	80.6	80	80.4	86.5	97.2	79.1	99.4	77.2	95	94.5	77	78.9	210007GXYL
	2	78.6	***	99.4	96.5	98.2	98.2	85.8	75.6	82.1	78.6	99.1	78.7	74.8	78.6	78	74.6	99.4	210008GDYX
	3	78.6	99.8	***	96.7	98.3	98.3	86	75.8	82.3	78.7	99.3	78.9	75	78.7	78.2	74.8	99.6	210018GXYL
	4	80	97.5	97.5	***	95.4	95.4	84.7	77.3	80.8	80.2	96.3	80.4	74.4	80	79.5	74.3	97	210021GXYL
	5	79.8	98.8	98.8	96.5	***	100	86.2	76	83.7	80.4	98	80.6	74.8	80.3	79.7	74.6	98.3	210024GXYL
	6	79.8	98.8	98.8	96.5	100	***	86.2	76	83.7	80.4	98	80.6	74.8	80.3	79.7	74.6	98.3	210031GXYL
	7	79.9	87.9	87.9	87	88.2	88.2	***	77.8	87.1	79.8	85.8	79.8	75.1	80	79.4	74.9	86	210059GXYL
	8	79.8	77	77	78.4	77.2	77.2	78.4	***	77.3	80.2	75.8	80.2	80	80.4	79.9	79.8	75.8	220198GDZC
	9	85.5	85.8	85.8	84.6	86.6	86.6	89.3	77.9	***	86.1	82.5	86.5	76.5	86.7	86.1	76.3	82.3	220149YNKM
	10	97.3	78.6	78.6	79.9	79.8	79.8	79.9	79.4	85.5	***	78.9	97.4	77	94.8	94.3	76.8	78.7	210085GXYL
	11	78.7	99.7	99.8	97.4	98.7	98.7	87.9	77.1	86	78.7	***	79.1	75.4	78.9	78.4	75.2	99.3	210127GXYL
	12	99.4	78.7	78.7	80.2	80	80	80	79.7	85.6	97.4	78.9	***	76.8	94.8	94.3	76.6	78.9	220150GXQZ
	13	77.7	78.6	78.6	77.8	78.2	78.2	78.4	80.6	79.3	77.9	78.7	77.7	***	77.7	77.2	99.6	75	220161YNKM
	14	95.1	78.4	78.5	79.9	79.6	79.6	79.8	80.3	85.4	94.6	78.6	95.3	77.7	***	99.3	77.6	78.7	210098GXNN
	15	95.1	78.3	78.4	79.8	79.5	79.5	79.6	80.2	85.2	94.5	78.5	95.1	77.4	99.6	***	77	78.2	210099GXNN
	16	77.6	78.5	78.5	77.7	78	78	78.4	80.6	79.2	77.7	78.6	77.6	99.9	77.6	77.2	***	74.8	FJ888351_H120 vaccine
	17	78.6	99.8	99.9	97.7	98.8	98.8	87.9	77	85.8	78.6	99.8	78.7	78.6	78.5	78.4	78.5	***	KF377577_4/91vaccine
		1	2	3	4	5	6	7	8	9	10	11	12	13	14	15	16	17	

**Table 3 viruses-15-01273-t003:** Antigenic correlation between vaccine strains and isolated strains.

	H120	4/91	210059GXYL	220161YNKM	210127GXYL	220149YNKM	210085GXYL	220198GDZC	210099GXNN
H120	100	6.54	2.21	74.56	10.06	3.41	4.71	27.6	14.12
4/91	2.1	100	9.94	5.33	63.41	11.12	6.57	8.766	9.44
210059GXYL	3.21	21.7	100	1.05	10.49	80.03	4.21	4.25	5.46
220161YNKM	54.36	3.62	2.44	100	14.52	11.35	12.96	21.31	15.23
210127GXYL	16.11	64.43	15.46	5.1	100	6.82	3.22	10.07	9.08
220149YNKM	1.75	18.87	59.86	4.87	6.82	100	5.97	5.41	4.12
210085GXYL	2.76	4.43	4.21	12.96	12.34	2.78	100	16.72	35.78
220198GDZC	35.1	17.58	3.67	18.65	4.62	5.41	16.72	100	17.46
210099GCYL	37.21	21.2	5.46	7.61	9.08	4.12	24.71	17.46	100

## Data Availability

The genomic data presented in this study are available from GenBank (accession numbers: OQ790027-OQ790033, OQ790035-OQ790037, OQ790043-OQ790048).

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
