# Peer review of "Antigenic Characterization of Infectious Bronchitis Virus in the South China during 2021–2022"

_viruses, 2023, doi:10.3390/v15061273_

Round 1

Reviewer 1 Report

Material&methods should be improved, Scientific papers should have a clear description of methods and study designs applied so as Manuscript needs minor revision especially in terms of material&methods and results description to improve transparency.

Manuscript needs major revision especially in terms of material&methods and results description to improve transparency.

Only on Fig 5 it became clear for the reviewer (and the future reader, too) which "isolated strains" were used for for chicken immunisation in oil-adjuvated form. It must be clearly indicated in the text, too including Material&methods and results, respectively.

It should also be provided a clear rationale, why you chose those isolated strains for immunisation study. 

On Fig 1 the presented information is not clear, as you had 16 inoculated groups, please indicate in a way for instance "study schedule A for group1-7".

For this reason I suggest revision of the text!

Author Response

Dear Reviewer #1:

Re: Manuscript ID: viruses-2405000

Thanks very much for taking your time to review this manuscript. I really appreciate all your comments and suggestions! Please find my itemized responses in below and my revisions/corrections in the re-submitted files.

Response to Reviewer #1

Comment 1: Only on Fig 5 it became clear for the reviewer (and the future reader, too) which "isolated strains" were used for for chicken immunisation in oil-adjuvated form. It must be clearly indicated in the text, too including Material&methods and results, respectively.

Response 1:Thanks for your suggestion. We have added information such as description of organizational samples (line 95 - 98), identification primers for IBV in the 'Virus isolation and identification' section (line 104 - 107), Cloning and Sequencing of S1 (line 108 - 121) and information about the reagents (line 146 and 148).

Comment 2: It should also be provided a clear rationale, why you chose those isolated strains for immunisation study.

Response 2: We selected these 7 strains from genotype and isolation sites, among which 210059GXYL, 220198GDZC, 220149YNKM, and 210099GXNN are potential strains for S1 gene recombination, 210085GXYL is a GI-19 strain isolated from Guangxi, 210127GXYL is a GI-13 strain isolated from Guangxi, and 220161YNKM is the only GI-1 strain isolated from Yunnan in this study. We have added this description in the (line 290 - 294).

Comment 3: On Fig 1 the presented information is not clear, as you had 16 inoculated groups, please indicate in a way for instance "study schedule A for group1-7".

Response 3: We have added grouping information for immune and infectious strains in Figure 1.

Reviewer 2 Report

Yuan and the colleagues isolated 15 IBV isolates from chickens with apparent symptoms, which were reproduced when challenged in SPF chickens. Genetic characterization of S1 gene revealed that the most prevalent was that of QX-like viruses from GI-19 lineage. Further investigation revealed recombination in S1 gene. Antigenic characterization by cross-serum neutralization revealed that no cross neutralization was observed between the selected seven strains possibly due to recombination. Authors then emphasize the need to match the vaccine strain, as genotype and serotype do not match. However, the authors’ claim that they have discovered a new trend of IBV prevalence is farfetched. The manuscript could be greatly improved by facilitating the editing services and including some key information that will help the readers to understand the manuscript more easily. Also, tables are not completely visible.

Here are some suggestions to the authors to improve the manuscript.

Line 83: “Virus isolation and identification” part of the M&M seems to be dramatically reduced, leaving some critical information out of the manuscript.

Line 84: authors should specify which tissue samples were used for the isolation of IBV.

Which also connects to the comment made below for line 250

Line 85: Authors should add details on how farms were chosen in southern China and specifically which regions in southern China. It would be beneficial for readers to find out about the positive rate of the surveillance in each region.

What kind of measures were taken to account for surveillance bias?

Line 86: 4/91 of GI-13 lineage is not Mass type

Line 87: omit “were”

Line 89: clarify which stage the “keeping in 4*C for 3h” was in the blind passaging process and why would it be necessary?

Line 93-94: please include a reference of the primers used in this study. If they were designed in this study please provide which sequences were used to design the primers, using which program of pcr, and the target size of the pcr product to improve the reproducibility of the paper. Also, cite any relevant reagents used in the process.

Line 96: authors should include the methods describing the sequencing of S1 gene to improve the reproducibility of this study.

Line 101: cite figtree

Line 113: omit “SPF”

Line 115-117:  this sentence should be rewritten to enhance the readability of the sentence for the readers. 

Figure 1: It is not clear which strains were picked for the animal experiment although available in Figure 5. Also, it is not clear why the strains were picked, out of the 15 viruses isolated in this study.

Line 119: please cite the information for the reagent.

Line 132: …”immunization”…

Figure 2. . Recommendation to the authors is to further genetically characterize GI-19 strains as they have been the most prevalent strain in China for a very long time now. There must be genetic diversity among the GI-19 lineages. Please characterize the GI-19 strains in to sub-lineages, genotypes as the authors see fit.

Line 177: typo “210021GXYL” should be “210059GXYL”

Line 194-195: What is the significance of this information? (table 3.) It is not mentioned again anywhere in the manuscript and seems irrelevant to the information that the authors is trying to portray. Also, if they were all well below 5.5 ( which is the inactivated vaccine dose in the animal experiment), how were the inactivated vaccines prepared with such low titer of viruses. Please include the method if they went through any concentrating procedure.

Line 201: typo “chichens”

Line 205: authors only briefly mention that the symptoms from the challenged chickens were consistent to that of the source-diseased chickens. How many of the chickens in a group exhibited the same symptoms? Authors should include the details of the clinical symptom observation post-challenge. Recommend using a clinical scoring system.

Line 219: If the homologous neutralization titer reached 64 and the VN titer of H120 and 4/91 against 220161YNKM and 210127GXYL has reached the titer of 32, one would not characterize it as poor neutralizing ability. Please explain the rationale.

Line 232-234: To our knowledge, researchers in China have developed multiple GI-19 lineage live-attenuated IBV vaccines. Authors should include this information and consider whether they should have included this into their vaccine study. If not, explain why no vaccines with matching genotype commercially available.

Line 247-252: Authors explain that the presumably low isolation rate (or non-IBV samples with respiratory symptoms) might be due to other pathogens exhibiting similar symptoms or co-infection. Our suggestion to the authors is to follow up on the samples to reveal(at least detection using RT-PCR or detection from AF if they were all inoculated in ECE) the prevalence of other pathogens in the rest of the samples. The connection of isolates and the clinical type without this process is unconvincing to the readers. Which then brings me to the results of  challenge study (in line 205) that only mentions that the symptoms were consistent to that of the source-diseased chickens.

Line 250-251: if so, should the authors try extracting RNA directly from the clinical samples and test the samples directly from the clinical samples.

 Line 283: what would constitute as “popular strains”

Line 289-291, 293: the study described in the 289-291 used whole genome of the IBV to characterize the genotype. The authors stopped at only sequencing S1. The disparity between the genotype of S1 and the serotype of the whole virus in an animal experiment could be from the rest of the genome, as S1 gene only makes up about 6% of the whole IBV genome and increasingly more recent studies suggesting that S1 is not enough to characterize the “serotype” or “protectotype”. Please address this issue in the manuscript with more references.

Table 1: genotype revealed by the sequencing of S1 gene of the isolates should be included in the table.

Line 325: numbers should be replaced with strain names to enhance the readability. Maybe include genotype in the column heading.

Recommendation for the authors is to receive a thorough english editing services provided by professional editing services.

Author Response

Dear Reviewer #2:

Re: Manuscript ID: viruses-2405000

We feel great thanks for your professional review work on our article. As you are concerned, there are several problems that need to be addressed. According to your nice suggestions, we have made extensive corrections to our previous draft, the detailed corrections are listed below.

Response to Reviewer #2

Comment 1: Line 83: “Virus isolation and identification” part of the M&M seems to be dramatically reduced, leaving some critical information out of the manuscript.

Response 1: Thanks for your suggestion. We have added information such as identification primers for IBV in the 'Virus isolation and identification' section (line 105 - 107).

Comment 2: Line 84: authors should specify which tissue samples were used for the isolation of IBV.Which also connects to the comment made below for line 250

Response 2: Thank you for pointing this out. We have added a description of organizational samples in this section (line 95 - 98).

Comment 3: Line 85: Authors should add details on how farms were chosen in southern China and specifically which regions in southern China. It would be beneficial for readers to find out about the positive rate of the surveillance in each region. What kind of measures were taken to account for surveillance bias?

Response 3: Thank you for pointing this out. The selected farms are those that have reported suspected IBV outbreaks to us, located in Yunnan, Guizhou, Guangxi, and Guangdong provinces in southern China. We have supplemented this information in our materials and methods (line 95 - 98, 268 - 271). Due to the varying levels of biosafety in aquaculture farms in different provinces and the large difference in sample numbers, monitoring bias is also an inevitable challenge.

Comment 4: Line 86: 4/91 of GI-13 lineage is not Mass type

Response 4: We have corrected the errors in the (line 98).

Comment 5: Line 87: omit “were”

Response 5: We feel sorry for our carelessness. Thanks for your correction.

Comment 6: Line 89: clarify which stage the “keeping in 4℃ for 3h” was in the blind passaging process and why would it be necessary?

Response 6: Usually, keeping in 4 ℃ for 3 hours is performed after the death of chicken embryos after inoculation. Reference: “Feng K, Xue Y, Wang F, et al. Analysis of S1 gene of avian infectious bronchitis virus isolated in southern China during 2011-2012. Virus Genes. 2014 Oct;49(2):292-303. doi: 10.1007/s11262-014-1097-1.”

Comment 7: Line 93-94: please include a reference of the primers used in this study. If they were designed in this study please provide which sequences were used to design the primers, using which program of pcr, and the target size of the pcr product to improve the reproducibility of the paper. Also, cite any relevant reagents used in the process.

Response 7: Thank you for your sincere suggestions. We have added this description in the Materials and Methods section (line104 -107), The primers used for RT-PCR are listed in Supplementary Tables S1.

Comment 8: Line 96: authors should include the methods describing the sequencing of S1 gene to improve the reproducibility of this study.

Response 8: We agree with the reviewer’s assessment. We have added this description in the Materials and Methods section (line 108 - 121).

Comment 9: Line 101: cite figtree

Response 9: We have cited the website of Figtree in this section (line 129).

Comment 10: Line 113: omit “SPF”

Response 10: We were really sorry for our careless mistakes. Thank you for your reminder.

Comment 11: Line 115-117: this sentence should be rewritten to enhance the readability of the sentence for the readers.

Response 11: In order to avoid ambiguity among readers, we have deleted this sentence without affecting the reading.

Comment 12: Figure 1: It is not clear which strains were picked for the animal experiment although available in Figure 5. Also, it is not clear why the strains were picked, out of the 15 viruses isolated in this study.

Response 12: We have added grouping information for immune and infectious strains in Figure 1. We selected these 7 strains from genotype and isolation sites, among which 210059GXYL, 220198GDZC, 220149YNKM, and 210099GXNN are potential strains for S1 gene recombination, 210085GXYL is a GI-19 strain isolated from Guangxi, 210127GXYL is a GI-13 strain isolated from Guangxi, and 220161YNKM is the only GI-1 strain isolated from Yunnan in this study. We have added this description in the (line 290 - 294).

Comment 13: Line 119: please cite the information for the reagent.

Response 13: In this section, we have added information about the reagents (line 146 and 148).

Comment 14: Line 132: …”immunization”…

Response 14: Thank you, we have changed “immunize” into “immunization”

Comment 15: Figure 2. Recommendation to the authors is to further genetically characterize GI-19 strains as they have been the most prevalent strain in China for a very long time now. There must be genetic diversity among the GI-19 lineages. Please characterize the GI-19 strains in to sub-lineages, genotypes as the authors see fit.

Response 15: Thank you for your interest in this part. At present, there is no international classification standard for the GI-19 subtype. We have analyzed all GI-19 strains submitted on Genbank and the GI-19 strains we have isolated in recent years. The research results in this section may affect the publication of subsequent articles, so we have not presented them in this article.

Comment 16: Line 177: typo “210021GXYL” should be “210059GXYL”

Response 16: We feel sorry for our carelessness. In our resubmitted manuscript, the typo is revised. Thanks for your correction.

Comment 17: Line 194-195: What is the significance of this information? (table 3.) It is not mentioned again anywhere in the manuscript and seems irrelevant to the information that the authors is trying to portray. Also, if they were all well below 5.5 (which is the inactivated vaccine dose in the animal experiment), how were the inactivated vaccines prepared with such low titer of viruses. Please include the method if they went through any concentrating procedure.

Response 17: We agree with your point of view that this information is irrelevant for follows-up research, so we have removed the description in this section and added virus concentration information to the method (line 145 ).

Comment 18: Line 201: typo “chichens”

Response 18: Thanks for your careful checks. We are sorry for our carelessness. In our resubmitted manuscript, the typo is revised.

Comment 19: Line 205: authors only briefly mention that the symptoms from the challenged chickens were consistent to that of the source-diseased chickens. How many of the chickens in a group exhibited the same symptoms? Authors should include the details of the clinical symptom observation post-challenge. Recommend using a clinical scoring system.

Response 19: Thank you for your suggestion. We have added the details of post challenge clinical symptom observation in the Supplementary Table S3

Comment 20: Line 219: If the homologous neutralization titer reached 64 and the VN titer of H120 and 4/91 against 220161YNKM and 210127GXYL has reached the titer of 32, one would not characterize it as poor neutralizing ability. Please explain the rationale.

Response 20: We believe that H120 and 4/91 have a VN titer of 32 against 220161YNKM and 210127GXYL, which only demonstrates stronger than other strains, but does not highlight its well neutralizing ability.

Comment 21: Line 232-234: To our knowledge, researchers in China have developed multiple GI-19 lineage live-attenuated IBV vaccines. Authors should include this information and consider whether they should have included this into their vaccine study. If not, explain why no vaccines with matching genotype commercially available.

Response 21: We agree that this is a potential limitation of the study. There are currently multiple GI-19 series IBV attenuated live vaccines available, but during our sampling process, we found that they have not been widely used. The diseased chickens were only specifically immunized with H120 or 4/91 vaccines. Therefore, our research also suggests the need for immunization against epidemic strains to contribute to IBV prevention and control in production. In subsequent studies, we will also investigate the protective effect of the GI-19 series IBV attenuated live vaccine on isolated virus strains. (line 321 -325)

Comment 22: Line 247-252: Authors explain that the presumably low isolation rate (or non-IBV samples with respiratory symptoms) might be due to other pathogens exhibiting similar symptoms or co-infection. Our suggestion to the authors is to follow up on the samples to reveal (at least detection using RT-PCR or detection from AF if they were all inoculated in ECE) the prevalence of other pathogens in the rest of the samples. The connection of isolates and the clinical type without this process is unconvincing to the readers. Which then brings me to the results of challenge study (in line 205) that only mentions that the symptoms were consistent to that of the source-diseased chickens.

Response 22: We have been conducting tests for other pathogens, including NDV, AIV, ALV, MDV, FADV, and MS, during the process of sample acquisition and virus isolation. We have indeed found cases of co infection with multiple pathogens. But we removed co infected samples and only isolated IBV strains from the isolated infection group, ensuring the presence of only IBV during the process. We have provided additional explanations in the materials and methods, results, and discussion sections (line 104-107, 169 and 268).

Comment 23: Line 250-251: if so, should the authors try extracting RNA directly from the clinical samples and test the samples directly from the clinical samples.

Response 23: We directly tested IBV from clinical samples, but there were co-infection with other viruses, which greatly affected the isolation of IBV. We did not purify the IBV in the co-infected samples, but only isolated 15 IBV strains from the samples infected with IBV alone (line 104-107, 169 and 268).

Comment 24: Line 283: what would constitute as “popular strains”

Response 24: In order to avoid ambiguity among readers, we have modified “popular trains” to “isolate trains”

Comment 25: Line 289-291, 293: the study described in the 289-291 used whole genome of the IBV to characterize the genotype. The authors stopped at only sequencing S1. The disparity between the genotype of S1 and the serotype of the whole virus in an animal experiment could be from the rest of the genome, as S1 gene only makes up about 6% of the whole IBV genome and increasingly more recent studies suggesting that S1 is not enough to characterize the “serotype” or “protectotype”. Please address this issue in the manuscript with more references.

Response 25: This study also concluded that the genotype represented by the S1 gene is not sufficient to represent serotype classification. We made corresponding modifications and added a references in the discussion section (line 317-321).

Comment 26: Table 1: genotype revealed by the sequencing of S1 gene of the isolates should be included in the table.

Response 26: We have added this section of information to the Table1.

Comment 27: Line 325: numbers should be replaced with strain names to enhance the readability. Maybe include genotype in the column heading.

Response 27: The rightmost column in the table contains the names of the strains, which may not have been displayed due to editing reasons. Therefore, we have also made adjustments.